https://doi.org/10.1038/s41467-020-17092-w | OPEN

# Dually modulated photonic crystals enabling high-power high-beam-quality two-dimensional beam scanning lasers

Ryoichi Sakata [1,2], Kenji Ishizaki[1,2], Menaka De Zoysa[1,2], Shin Fukuhara[1], Takuya Inoue [1], Yoshinori Tanaka[1], Kintaro Iwata[1], Ranko Hatsuda[1], Masahiro Yoshida [1], John Gelleta [1] & Susumu Noda [1✉]

Mechanical-free, high-power, high-beam-quality two-dimensional (2D) beam scanning lasers are in high demand for various applications including sensing systems for smart mobility, object recognition systems, and adaptive illuminations. Here, we propose and demonstrate the concept of dually modulated photonic crystals to realize such lasers, wherein the positions and sizes of the photonic-crystal lattice points are modulated simultaneously. We show using nano-antenna theory that this photonic nanostructure is essential to realize 2D beam scanning lasers with high output power and high beam quality. We also fabricate an on-chip, circuit-driven array of dually modulated photonic-crystal lasers with a 10 × 10 matrix configuration having 100 resolvable points. Our device enables the scanning of laser beams over a wide range of 2D directions in sequence and in parallel, and can be flexibly designed to meet application-specific demands.

[1] Department of Electronic Science and Engineering, Kyoto University, Kyoto 615-8510, Japan. [2] These authors contributed equally: Ryoichi Sakata, Kenji Ishizaki, Menaka De Zoysa. ✉email: snoda@kuee.kyoto-u.ac.jp

Two-dimensional (2D) beam scanning is an essential technique for various applications including light detection and ranging (LiDAR) systems in autonomous driving[1–4] and smart mobility of robots[5], object recognition systems[6], and adaptive illuminations[7]. Most beam-scanning methods today rely on steerable mechanical components and thus suffer from poor reliability and stability, low operation speeds, and bulky device sizes. As an answer to these deficiencies, phased arrays[8–14] based on silicon photonics recently have been attracting attention, particularly for LiDAR applications. However, the most advanced fully 2D phased arrays suffer from a narrow field of view of less than ~5° and the presence of many grating lobes[9]. Thus, most phased arrays for 2D beam steering combine two mechanisms of the phased array itself and a grating diffraction based on wavelength tuning (150–200 nm) of an external light source[14]. Here we note that there are two schemes for LiDAR[15]: (a) a time-of-flight (ToF) scheme and (b) a coherent scheme involving a frequency-modulated continuous-wave (FMCW) method. Between these, the ToF scheme is mostly utilized for autonomous driving and robot mobility owing to its simplicity; this scheme requires a high-peak output power (watt-class or even higher, depending on the desired range of distances) and a fixed narrow emission bandwidth so that a narrow-bandpass filter can be utilized to remove the background noise of solar light, etc. Phased arrays, whose output power is limited by two-photon absorption to at most a few tens of milliwatts when made of silicon, and/or whose wavelength bandwidth is required to be as wide as 150–200 nm, struggle to satisfy these requirements and thus are difficult to be applied to the ToF scheme. Rather, phased arrays are mainly considered for the coherent FMCW scheme, which does not require high-output power nor a fixed, narrow emission wavelength, yet in exchange involves much more complicated implementations. If we could realize a new, nonmechanical 2D beam-scanning method applicable to even a ToF scheme, then we could offer a greater variety of choices in the sensing fields for autonomous driving and robot mobility, contributing to the progress of smart Society 5.0 (proposed by the Japanese government as a future society[16]). This nonmechanical, high-power, high-beam-quality 2D beam-scanning technique would be also useful to other applications, such as adaptive illumination, etc.

A promising approach to satisfy the above requirements is to develop compact, on-chip, 2D beam-scanning semiconductor lasers. Toward this goal, initial studies on these lasers have been performed using 2D photonic crystals[17,18]. In particular, the feasibility of emitting a beam in a desired direction in two dimensions has been investigated using tailored 2D photonic-crystal structures[18]. However, fundamental problems, to be detailed in the next section, have been found to impede the realization of ideal beam-scanning lasers with high-output power and high-quality beams.

In this paper, we propose the concept of dually modulated photonic crystals, and show that the simultaneous modulation of both the positions and sizes of the photonic-crystal lattice points are essential to realize the emission of a beam in any 2D direction with a high-output power and high-quality beam. This concept is developed based on nanoantenna theory and demonstrated experimentally. We fabricate arrays of 100 dually modulated photonic-crystal lasers in a $10 \times 10$ matrix configuration, which can be operated by high-speed, circuit-based switching, to demonstrate 2D beam scanning, as well as the simultaneous scanning of multiple beams, over a wide range of directions. Our concept will allow the number of resolvable points, divergence angle, output power, and total device size to be flexibly modified to meet application-specific targets, as described later. With such excellent performance and novel functionalities, our devices will offer a greater variety of choices in sensing fields for autonomous

driving and robot mobility. This nonmechanical, high-power, high-beam-quality 2D beam-scanning technique is also useful to applications including adaptive illuminations.

## Results

### Proposal of dually modulated photonic crystals.
A schematic structure of our proposed dually modulated photonic crystal is illustrated in Fig. 1a. The positions and sizes of the lattice points of this structure are simultaneously modulated with respect to the original structure shown in Fig. 1b. For lasing oscillation, we make use of band-edge resonance at the $M_1$ high-symmetry point of the photonic crystal, highlighted with a red circle in the photonic band diagram in Fig. 1c. At this point, laser oscillation is sustained at four band edges—A, B, C, and D (see the inset of Fig. 1c)—by the direct coupling of four fundamental waves —$R_1$, $R_2$, $R_3$, and $R_4$—propagating in the crystal via single reciprocal lattice vectors (e.g., $G_{1,1}$) and the indirect coupling of these waves via a combination of two reciprocal lattice vectors (e.g., $G_{1,1}$ and $G_{-1,0}$) through high-order Bloch waves; these couplings are summarized in Fig. 1d. Note that, at the $M_1$-point, waves $R_1 - R_4$ lie outside the air-light cone, and thus the lasing light cannot be emitted to free space. This is different from the surface-emitting photonic-crystal lasers reported elsewhere[19,20], for which the $\Gamma_2$-point is used for lasing oscillation, leading to the emission of a beam in a direction normal to the surface of the photonic crystal.

Next, we modulate the photonic-crystal lattice points as shown in Fig. 1a. The key innovation here is the modulation of not only the position of these lattice points, which was also done in previous work[18], by

$$\bar{\mathbf{r}}_{m,n} = \mathbf{r}_{m,n}^0 + \Delta\mathbf{d}\sin(\mathbf{k} \cdot \mathbf{r}_{m,n}^0) \tag{1}$$

but also their size, as

$$\bar{S}_{m,n} = S_0 + \Delta S\sin(\mathbf{k} \cdot \mathbf{r}_{m,n}^0). \tag{2}$$

Here, $\bar{\mathbf{r}}_{m,n}$ ($\mathbf{r}_{m,n}^0$) are the position vectors of the lattice points after (before) modulation, $\Delta\mathbf{d}$ is the position modulation vector, $\bar{S}_{m,n}$ ($S_0$) are the sizes of lattice points after (before) modulation, and $\Delta S$ is the amplitude of size modulation.

In Eqs. (1) and (2), $\mathbf{k}$ denotes a diffraction vector which diffracts the four fundamental waves $R_1$, $R_2$, $R_3$, and $R_4$) inside the air-light cone, leading to the generation of vector $\mathbf{K}$, which determines the emission direction in free space given by polar angle $\theta$ and azimuthal angle $\phi$ (see Fig. 1d), where $\mathbf{K} = \left(\frac{2\pi}{\lambda}\right)(\sin\theta\cos\phi, \sin\theta\sin\phi)$ and $\lambda$ is the wavelength in free space. We note that for the generated vector $\mathbf{K}$, both positive and negative directions ($\mathbf{K}$ and $-\mathbf{K}$) are possible because the modulation given by Eqs. (1) and (2) induces not only positive but also negative $\mathbf{k}$ vectors (see Fig. 1d). Thus, the emission occurs simultaneously in two directions, $(\theta, \phi)$ and $(\theta, \phi + 180°)$. Here, we should note that the concept proposed here is in no way based on "diatomic" structures or designs, which were applied, for example, in refs. [17,20], where two photonic crystals are structurally overlapped and the interaction between pairs of photonic atoms in the combined crystal plays an important role (see Supplementary Note 1). We also emphasize that the present device is completely different from a simple combination of independently fabricated 2D grating and surface-emitting photonic-crystal lasers (or vertical cavity surface-emitting lasers), as detailed in Supplementary Note 2.

Next, we describe why our dually modulated photonic crystal is important for realizing ideal beam-scanning lasers, using nanoantenna theory. Here, we provide an intuitive, qualitative explanation; a quantitative, rigorous explanation is given in Supplementary Note 3. Figure 2a shows the electric-field

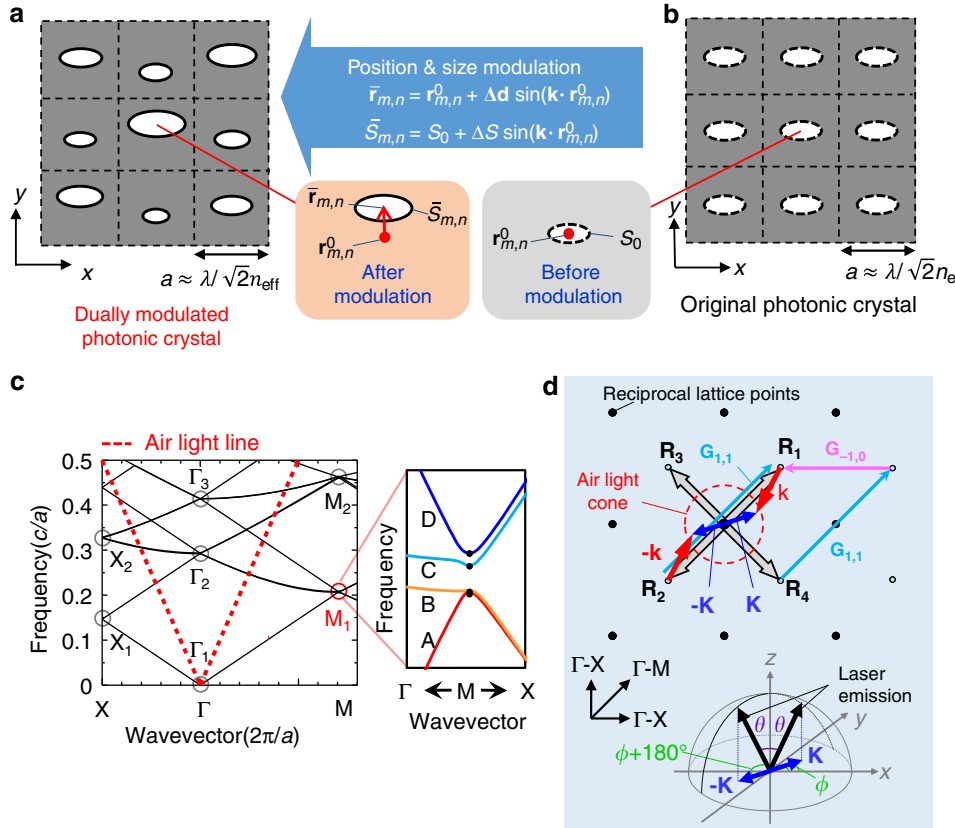

**Fig. 1 Dually modulated photonic crystal for laser beam scanning. a**, **b** Schematic diagram of dual modulation. **a** Dually modulated photonic crystal whose lattice-point position vectors $\bar{\mathbf{r}}_{m,n}$ and sizes $\bar{S}_{m,n}$ have been simulaneously modulated from their original values $\mathbf{r}^0_{m,n}$ and $S_0$ by amounts $\Delta \mathbf{d} \cdot \sin(\mathbf{k} \cdot \mathbf{r}^0_{m,n})$ and $\Delta S \cdot \sin(\mathbf{k} \cdot \mathbf{r}^0_{m,n})$, respectively, where integers $m$ and $n$ specify the lattice point, $\Delta \mathbf{d}$ is the position modulation vector, $\Delta S$ is the size modulation amplitude, and $\mathbf{k}$ is a diffraction vector that diffracts the four fundamental waves ($\mathbf{R}_1, \mathbf{R}_2, \mathbf{R}_3$, and $\mathbf{R}_4$ in **d**) inside the air light cone. $a$ is the photonic-crystal lattice constant and $\lambda$ is the wavelength in free space. **b** The original photonic crystal before modulation. **c** Band diagram of a square-lattice photonic crystal, and a magnified view of the four band-edges A, B, C, and D at the $M_1$-point, which lies below the air light line and is used for lasing. **d** Schematic diagram of light diffraction in reciprocal lattice space at the $M_1$-point. Diffraction of the four fundamental waves ($\mathbf{R}_1, \mathbf{R}_2, \mathbf{R}_3$, and $\mathbf{R}_4$) by $\mathbf{k}$ leads to the generation of vector $\mathbf{K}$, where $\mathbf{K} = \left(\frac{2\pi}{\lambda}\right)(\sin\theta\cos\phi, \sin\theta\sin\phi)$ with the emission direction in free space given by polar angle $\theta$ and azimuthal angle $\phi$ and the wavelength in free space given by $\lambda$. Note that both positive and negative vectors ($\mathbf{K}$ and $-\mathbf{K}$) are generated because the modulation induces not only positive but also negative $\mathbf{k}$ vectors. Thus the emission occurs simultaneously in two directions, ($\theta, \phi$) and ($\theta, \phi + 180°$). $\mathbf{G}_{1,0}$ and $\mathbf{G}_{1,1}$ are reciprocal lattice vectors.

distributions of band-edge modes A, B, C, and D in the plane of the original, unmodulated photonic crystal. (A derivation of these electric fields are provided in Supplementary Note 3 and Supplementary Fig. 5.) Here, let us regard the individual lattice points as nanoantennas, which radiate the electric field to free space. In the far field, this radiated electric field, $\mathbf{E}_{\text{far}}(\mathbf{K})$, is determined by the Fourier transform of the electric-field distribution, $\mathbf{E}_{\text{aperture}}(\mathbf{r})$, inside every nanoantenna, and is expressed by (see the details in Supplementary Note 3, and Supplementary Eq. (1))

$$\mathbf{E}_{\text{far}}(\mathbf{K}) = \text{C} \iint dx dy \mathbf{E}_{\text{aperture}}(\mathbf{r}) \exp(i\mathbf{K} \cdot \mathbf{r}). \qquad (3)$$

The integrals in Eq. (3) are evaluated over every nanoantenna inside the photonic crystal and can be approximated as the following double sum:

$$\mathbf{E}_{\text{far}}(\mathbf{K}) \cong \text{C} \sum_m \sum_n [\bar{S}_{m,n} \mathcal{E}(\bar{\mathbf{r}}_{m,n})] \exp\left(i\mathbf{K} \cdot \bar{\mathbf{r}}_{m,n}\right). \qquad (4)$$

Equation (4) is valid because each nanoantenna is arranged discretely at position $\bar{\mathbf{r}}_{m,n}$ and the integral of $\mathbf{E}_{\text{aperture}}(\mathbf{r})$ can be approximated as the product of the size $\bar{S}_{m,n}$ of each nanoantenna

and the electric-field $\mathcal{E}(\bar{\mathbf{r}}_{m,n})$ at its center (see the details in Supplementary Note 3 and Supplementary Eq. (14)).

Returning to Fig. 2a, in the absence of modulation, the electric fields $[\bar{S}_{m,n} \mathcal{E}(\bar{\mathbf{r}}_{m,n})]$ ($= [S_0 \mathcal{E}(\mathbf{r}^0_{m,n})]$) in Eq. (4) at each nanoantenna can be regarded to be zero for band-edge modes A and B due to the rotational symmetry of their electric fields with respect to the center of each nanoantenna, as evident in the upper panels of Fig. 2a; thus, emission does not occur for band-edge modes A and B (see Supplementary Note 3 and Supplementary Eq. (15)). For band-edge modes C and D, the electric-field $[S_0 \mathcal{E}(\mathbf{r}^0_{m,n})]$ at each nanoantenna is finite (see the red and blue arrows in the lower panels of Fig. 2a); however, the electric-field vectors of adjacent nanoantennas point in opposite directions, so the total emission is cancelled out in the far field (see Supplementary Note 3 and Supplementary Eq. (17)). These results are consistent with the fact that the $M_1$-point lies outside the air-light cone and the lasing light cannot be emitted to free space, as described earlier.

Let us now introduce modulation. First, we introduce only position modulation[18]. The in-plane electric fields $[\bar{S}_{m,n} \mathcal{E}(\bar{\mathbf{r}}_{m,n})]$ ($= [S_0 \mathcal{E}(\bar{\mathbf{r}}_{m,n})]$) in Eq. (4) for the band-edge modes A, B, C, and D are shown in Fig. 2b (see the red and blue arrows). Here, we note that, among these four modes, lasing oscillation occurs in the mode of lowest loss, namely, the mode with the smallest far-field

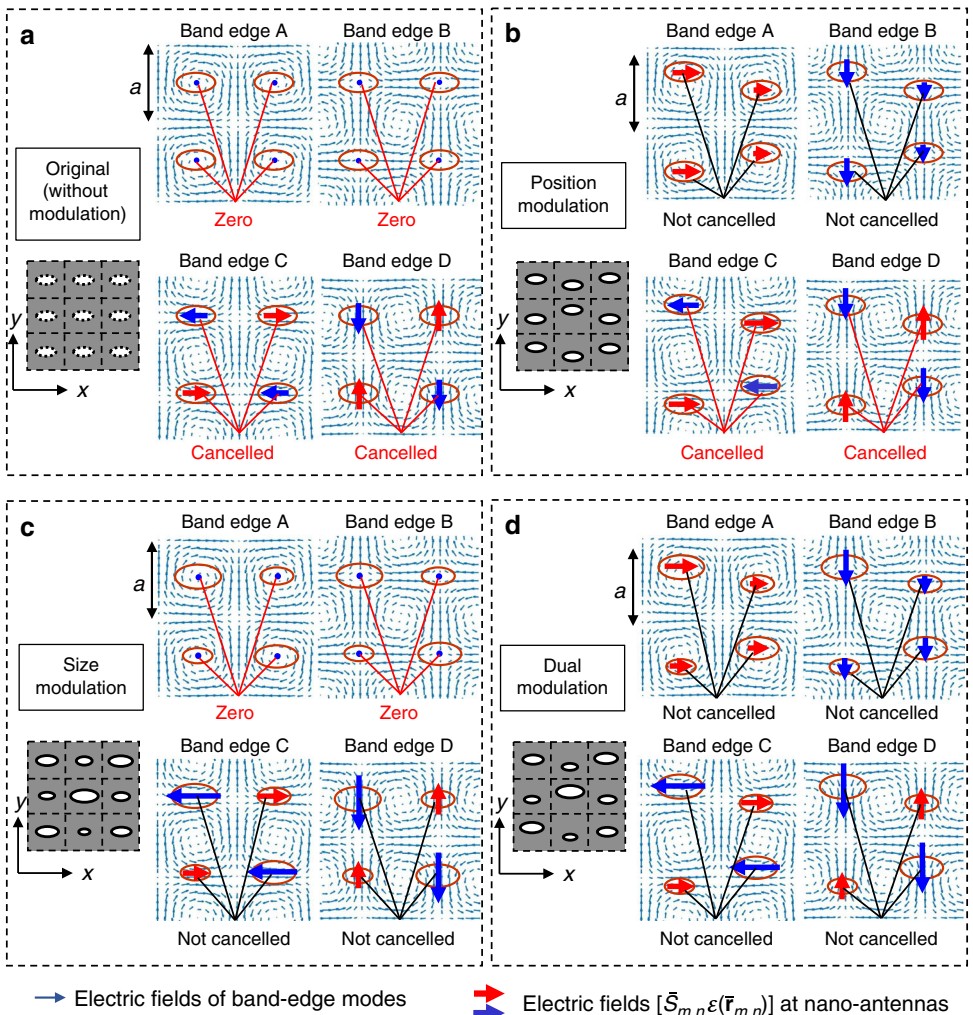

**Fig. 2 Elecric fields of band-edge modes A–D radiated into free-space by lattice points considered to function as nanoantennas. a–d** Electric-field distributions of band-edge modes A–D in a photonic crystal **a** without modulation and with **b** position modulation, **c** size modulation, and **d** dual modulation. **a** Far-field radiation is zero for all modes, due to the rotationally symmetric electric field of band-edge modes A and B with respect to the lattice points and the opposing electric-field vectors of band-edge modes C and D. **b** Far-field radiation is finite for band-edge modes A and B due to their finite electric-field vectors pointing in identical directions, and near-zero for band-edge modes C and D due to their finite electric-field vectors pointing in opposite directions. **c** Far-field radiation is zero for band-edge modes A and B, for the same reason as in **a**, and finite for band-edge modes C and D, due to the alleviation of destructive interference. **d** Far-field radiation is finite for all modes, due to the combined effects of position and size modulations. In each panel, the product of the size $\bar{S}_{m,n}$ of each nanoantenna and the electric-field $\mathcal{E}(\bar{\mathbf{r}}_{m,n})$ at its center position $\bar{\mathbf{r}}_{m,n}$ is shown with blue and red arrows. When the product becomes zero, only dots are shown.

emission; thus, proper control of the far-field emission of all band-edge modes is necessary. For band-edge modes A and B, the electric fields $[S_0\mathcal{E}(\bar{\mathbf{r}}_{m,n})]$ become finite at each nanoantenna, and their vectors point in the same direction, as evident in the upper panels of Fig. 2b. As a result, the far-field electric-field $\mathbf{E}_{\mathrm{far}}(\mathbf{K})$ can be expressed by the following equation (see Supplementary Note 3 and Supplementary Eq. (23)).

$$|\mathbf{E}_{\mathrm{far}}(\mathbf{K})| \propto |\mathbf{R}_j||\Delta\mathbf{d}| \propto |\Delta\mathbf{d}| \quad (5)$$

Equation (5) indicates that the far-field emission strength is proportional to the amplitude of position modulation $|\Delta\mathbf{d}|$. Thus, a sufficiently high far-field emission strength is expected for band-edge modes A and B. Meanwhile, for band-edge modes C and D, the electric fields inside the nanoantennas remain finite, but their vectors at adjacent nanoantennas continue to point in opposite directions, as evident in the lower panels of Fig. 2b. This once again leads to their destructive interference in the far field, although the destructive interference is not complete due to the

effect of position modulation. The resultant $\mathbf{E}_{\mathrm{far}}(\mathbf{K})$ is given by the following equation (see Supplementary Note 3 and Supplementary Eq. (28)).

$$|\mathbf{E}_{\mathrm{far}}(\mathbf{K})| \propto |\mathbf{K} \cdot \Delta\mathbf{d}| \quad (6)$$

$|\mathbf{E}_{\mathrm{far}}(\mathbf{K})|$ given by Eq. (6) is much smaller than that of Eq. (5) and can be regarded as essentially zero for the following reasons: When $\mathbf{K}$ and $\Delta\mathbf{d}$ are orthogonal, $|\mathbf{E}_{\mathrm{far}}(\mathbf{K})|$ in Eq. (6) becomes zero, and even when $\mathbf{K}$ and $\Delta\mathbf{d}$ are parallel, $|\mathbf{E}_{\mathrm{far}}(\mathbf{K})|$ in Eq. (6) is found to be much smaller than that in Eq. (5) because the amplitude of $|\mathbf{K}|$ is much smaller than $|\mathbf{R}_j|$ (see the details in Supplementary Note 3). Thus, lasing oscillation occurs at band-edge modes C or D instead of band-edge modes A and B, due to the much lower far-field emission strength (loss) of the former. This implies that position modulation alone, as given by Eq. (1), suffers from the fundamental problem of very low-output power. In addition, the cancellation of the far-field deteriorates the beam quality, and even causes emission-direction (**K**) dependence as

indicated in Eq. (6). These are the reasons why the initial study[18] is inherently problematic.

The outstanding issue of the above position modulation is the small far-field emission strength of band-edges modes C and D. We perceive that this issue could be solved by the introduction of size modulation. The in-plane electric fields of band-edge modes C and D at each nanoantenna $[\bar{S}_{m,n}\mathcal{E}(\bar{\mathbf{r}}_{m,n})]$ $(= [\bar{S}_{m,n}\mathcal{E}(\mathbf{r}^0_{m,n})])$ are illustrated in the lower panels of Fig. 2c (see the red and blue arrows). It is seen here that although the electric-field vectors of adjacent nanoantennas point in opposite directions as before, the amplitudes of these vectors are now different due to the effect of size modulation. Hence, the destructive interference of the electric fields emitted into the far field by adjacent nanoantennas is significantly alleviated. The resultant far-field emission is given as follows (see Supplementary Note 3 and Supplementary Eq. (34)).

$$|\mathbf{E}_{\text{far}}(\mathbf{K})| \propto |\Delta S|. \tag{7}$$

Equation (7) indicates that the far-field emission is proportional to the amplitude of size modulation $|\Delta S|$. Thus, a sufficiently high far-field emission strength is expected for band-edge modes C and D. However, we note that there now exists an issue for band-edge modes A and B instead: As indicated in the upper panels of Fig. 2c, $[\bar{S}_{m,n}\mathcal{E}(\mathbf{r}^0_{m,n})]$ is zero due to the same rotational electric-field symmetry that inhibited the far-field emission of these modes in unmodulated photonic crystals. Thus, their far-field electric-field $|\mathbf{E}_{\text{far}}(\mathbf{K})|$ remains zero (see Supplementary Note 3 and Supplementary Eq. (31)):

$$|\mathbf{E}_{\text{far}}(\mathbf{K})| = 0 \tag{8}$$

As a result of the vanishing far-field emission strength (i.e., loss) of band-edge modes A and B, lasing oscillation occurs at band-edge mode A or B instead of band-edge modes C or D. This implies that size modulation alone, as given by Eq. (2), also suffers from the fundamental problem of almost no output power.

By taking the best of the above two modulation schemes, we finally arrive at the idea of "dually modulated photonic crystals" as shown in Fig. 1a. With dual modulation, the in-plane electric fields $[\bar{S}_{m,n}\mathcal{E}(\bar{\mathbf{r}}_{m,n})]$ for band-edge modes A, B, C, and D are as shown in Fig. 2d. This figure clearly indicates that the nanoantennas work efficiently, and thus a reasonably large far-field emission strength is expected, for all band-edge modes. In particular, using a process similar to that used in the derivation of Eq. (5), the far-field emission strength of band-edge modes A and B is as follows (see Supplementary Note 3 and Supplementary Eq. (35)).

$$|\mathbf{E}_{\text{far}}(\mathbf{K})| \propto |\Delta\mathbf{d}|. \tag{9}$$

Meanwhile, using a process similar to that used in the derivation of Eq. (7), the far-field emission strength of band-edges modes C and D is as follows (see Supplementary Note 3 and Supplementary Eq. (37)).

$$|\mathbf{E}_{\text{far}}(\mathbf{K})| \propto |\Delta S| \tag{10}$$

Thus, we can expect a reasonably high far-field emission strength from all four band-edge modes, and we can choose between band-edge mode pairs (A, B) and (C, D) by controlling $|\Delta\mathbf{d}|$ and $|\Delta S|$. This realization of high far-field emission strength implies that dually modulated photonic-crystal structures (Fig. 1a) are very promising for realizing ideal beam-scanning lasers with high-output power and high beam quality. Herein lies the most important achievement of this work.

To supplement the qualitative analysis based on nanoantenna theory provided thus far, we next calculate the radiation constant $\alpha_v$ for each band-edge mode in the dually modulated photonic crystal using a three-dimensional coupled wave theory (3D-CWT)[21]. $\alpha_v$ of the lasing mode determines the slope efficiency and the output power of the laser. For these calculations, we extended 3D-CWT, which was originally formulated for the analysis of only unmodulated PC lasers operating at the $\Gamma_2$ point, to consider operation at the $M_1$-point[22] and include information relevant to diffraction vector $\mathbf{k}$ introduced by modulation. We consider here the radiation components only to free space, distinguished from that into other, unintended directions inside the device (i.e., radiation components into cladding layers and the substrate, which become loss). The results of these calculations were consistent with the above nanoantenna theory. We also confirmed that, in the dually modulated photonic crystals, the radiation components, which become loss are much smaller than that emitted in the desired direction in free space (see Supplementary Fig. 3).

Figure 3a shows $\alpha_v$ as a function of $|\Delta\mathbf{d}|$ when $|\Delta S| = 0.03a^2$, where $a$ is the lattice constant. Note that we set the direction of position modulation $\Delta\mathbf{d}$ to the $y$ direction as shown in Fig. 1a by considering the fabrication process as described in the next section. Other relevant device parameters are given in Supplementary Note 4. Figure 3b shows $\alpha_v$ as a function of $|\Delta S|$ when $|\Delta\mathbf{d}| = 0.08a$. Again the direction of position modulation $\Delta\mathbf{d}$ is set to the $y$ direction. These figures indicate that an appropriate selection of $|\Delta\mathbf{d}|$ and $|\Delta S|$ leads to a reasonably large radiation constant for any band-edge mode; that is, regardless of which band-edge mode is ultimately used for lasing, high-power, high beam quality operation can be assured. For example, when we select $|\Delta\mathbf{d}| = 0.08a$ and $|\Delta S| = 0.03a^2$, we obtain $\alpha_v \approx 17\,\text{cm}^{-1}$

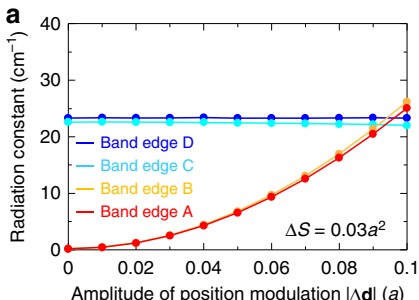
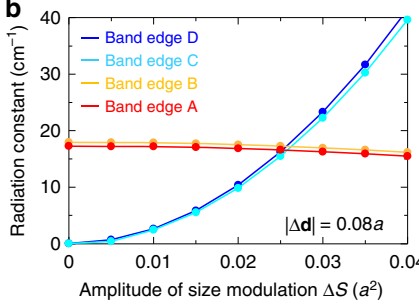

**Fig. 3 Calculated radiation constants for dually modulated photonic crystals by a three-dimensional coupled wave theory. a** Radiation constants of band-edges A–D with respect to the amplitude of position modulation $|\Delta\mathbf{d}|$, where the amplitude of size modulation $\Delta S$ is fixed to $0.03a^2$. **b** Radiation constants with respect to the amplitude of size modulation $\Delta S$, where the amplitude of position modulation $|\Delta\mathbf{d}|$ is fixed to $0.08a$. Throughout these calculations, the polar emission angle $\theta$ and azimuthal emission angle $\phi$ are fixed to $36°$ and $0°/180°$, respectively.

for band-edge modes A and B. In this case, even though we do not use the light emitted downward from the dually modulated photonic crystal, we can obtain a reasonably high slope efficiency of ~0.4 W/A (see Supplementary Note 4). If we would opt to use the downward-emitted light, by reflecting it upward with a distributed Bragg reflector (DBR), a doubled efficiency of 0.8 W/A and beyond is expected.

**Demonstration of dually modulated photonic-crystal lasers.** Based on the above concept, we construct an on-chip, compact 2D beam-scanning device, wherein we fabricate lasers with dually modulated photonic-crystal structures. In total, we integrate 100 lasers with different dually modulated photonic crystals onto a single chip in a $10 \times 10$ matrix configuration shown schematically in Fig. 4a. We have chosen this number of resolvable points for the purpose of providing sufficient proof of the usefulness of the concept of beam scanning over a wide range of polar angles and azimuthal angles, as detailed below. The lasers are integrated on a thick n-GaAs layer atop a semi-insulating (SI) GaAs substrate. Each laser is electrically isolated from the others in the form of a mesa with a surrounding isolation groove etched into the SI-GaAs substrate. Both p- and n-electrodes are formed onto the back side of the device (see Fig. 4a) and connected by a matrix of p- and n-line-electrodes, allowing for individual lasers at the intersection of these lines to be selectively driven. The laser light is emitted toward the front side, namely, from the surface of a SI-GaAs substrate with an anti-reflection layer.

To each photonic crystal within the $10 \times 10$ matrix, a dual modulation is applied such that a laser beam can be emitted over a wide range of polar angles ($0° \leq \theta \leq 45°$) and azimuthal angles ($0° \leq \phi \leq 180°$). Here, we should note that a corresponding twin beam with the same polar angle ($\theta$) but different azimuthal angle ($\phi + 180°$) is emitted simultaneously, as described earlier. Hereafter, we denote the azimuthal angle as $\phi/\phi + 180°$ to indicate the presence of the twin beam. Figure 4b shows the designed polar and azimuthal angles of the $10 \times 10$ matrix device. On lines p-1, p-2, p-3, p-6, p-7, and p-8, the azimuthal angles are fixed to $\phi = 0°/180°$ or $90°/270°$ and the polar angles are varied from $\theta = 0°$ to $45°$. On lines p-4, p-5, p-9, and p-10, the polar angles are fixed to $\theta = 10°$ or $20°$ and the azimuthal angles are varied from $\phi = 0°/180°$ to $180°/360°$. In order to check the fabrication consistency within the same device, pairs of lines (p-1/p-2 and p-6/p-7) are designed with the same angles.

The device was fabricated as follows. First, 5 μm of n-GaAs and the laser structure were grown on a SI-GaAs substrate by metal–organic vapour phase epitaxy (MOVPE). Next, dually modulated photonic crystals were formed into the GaAs layer by a combination of electron-beam lithography, dry etching, and air-hole-retained crystal regrowth[23]. Figure 5a shows a top SEM image (before regrowth) and cross-sectional SEM image (after regrowth) of the fabricated dually modulated photonic crystal. These images reveal that the dually modulated photonic-crystal patterns were well formed. Note that the direction of $\Delta \mathbf{d}$ was set to the y-direction as described in the previous section; otherwise the individual lattice points would overlap with each other, and the finished dually modulated photonic-crystal structure would be significantly deteriorated. Then, a mesa array and isolation grooves were etched by a combination of photolithography and dry etching. After this, p- and n-type contact and line electrodes were deposited. SiN$_x$ was then deposited as an anti-reflection layer on the surface of the SI-GaAs substrate by plasma-enhanced chemical vapor deposition. A more detailed description of the fabrication process is provided in Supplementary Note 5. We set the diameter of the p-electrode to 100 μm, which determines the

area of emission, and the mesa size to $150 \times 150$ μm², and arranged the mesas at a pitch of 265 μm in a $10 \times 10$ matrix. The size of the whole device was less than 3 mm. An optical microscope image of the back side of the finished device (with electrodes clearly visible) is shown in Fig. 5b. It is evident from this image that the matrix structure is well formed. Finally, the device was flip-chip bonded to a package as shown in Fig. 5c, so that each element could be operated by injecting the current via the metallic pins of the package. The laser beam was emitted from the surface of the SI substrate upon which the anti-reflection coating was deposited.

Following fabrication, we tested one of the 100 integrated lasers (at the intersection of electrodes n-1 and p-4 of Fig. 4b), which was designed to emit light for $\theta = 10°$ and $\phi = 0°/180°$. The current-output (I-L) characteristics of this laser element are shown in Fig. 5d. The slope efficiency was estimated to be ~0.4 W/A. This value agrees well with our theoretical prediction. As a result, the maximum power reached the order of one watt. Further improvement to the efficiency (>0.8 W/A) and power is expected by introducing a DBR to reflect the unused light that is emitted toward the bottom side of the device, as described in the previous section. The lasing wavelength was around 940 nm (specifically, ~949 nm when the oscillation occurs in the mode of band-edge A or B, and ~932 nm when in the mode of band-edge C or D). We have evaluated the standard deviation of these lasing wavelengths for all 100 samples, and found that the standard deviation from the mean wavelength is as small as around ~0.2 nm, which is within the resolution limit of our spectrometer. Because the temperature dependence of the lasing wavelength is ~0.086 nm/K, the required bandwidth, accommodating a temperature change of ~100 °C, could be less than 30 nm. The far-field pattern (FFP) of the device is shown in Fig. 5e. This figure shows that the element emitted a stable, single-lobed beam in the designed directions of $\theta = 10°$ and $\phi = 0°/180°$. The full-width-at-half-maximum (FWHM) divergence of the beam was confirmed to be nearly diffraction-limited (<1°). An even higher output power and lower beam-divergence angle are expected by reflecting the downward-emitted light upward and/or increasing the diameter of the p-electrode. We then measured the I-L characteristics of all other elements and confirmed that all had similar I-L characteristics; Fig. 5f is a color map of the slope efficiency of every element. The average slope efficiency was 0.42 W/A with a standard deviation of 0.04 W/A, in agreement with the slope efficiency of 0.4 W/A as designed. We also evaluated the average value and standard deviation of the FWHM beam-divergence angles; the average value and standard deviation were 0.7° and ~0.1°, respectively, where the standard deviation of these angles was found to be almost the same as the resolution of our measurement system. We also confirmed that the homogeneity of the beam emission directions for pairs of lines (p-1/p-2 and p-6/p-7), which were designed to emit beams in the same directions, are within the measurement resolution (<0.1°).

Finally, we performed 2D beam scanning by driving individual elements of the $10 \times 10$ matrix array, emitting beams over a wide range of polar angles (from $\theta = 0°$ to $45°$) and azimuthal angles (from $\phi = 0°/180°$ to $180°/360°$). The switching of individual elements was controlled by a microcontroller. We note that our device did not require a wavelength-tunable external laser source nor a temperature-induced refractive index change for beam scanning. Figure 6a, b shows several snapshots of 2D beam scanning with the fabricated device. In the first and second rows of Fig. 6a, we performed the scanning of polar angles (from $\theta = 0°$ to $45°$) with fixed azimuthal angles $\phi = 0°/180°$ or $90°/270°$ by driving one p-line electrode (p-1 or p-6) and switching the n-line electrodes (from n-1 to n-10). As indicated by the figure panels, single-lobed beams can be obtained in any designed direction. In

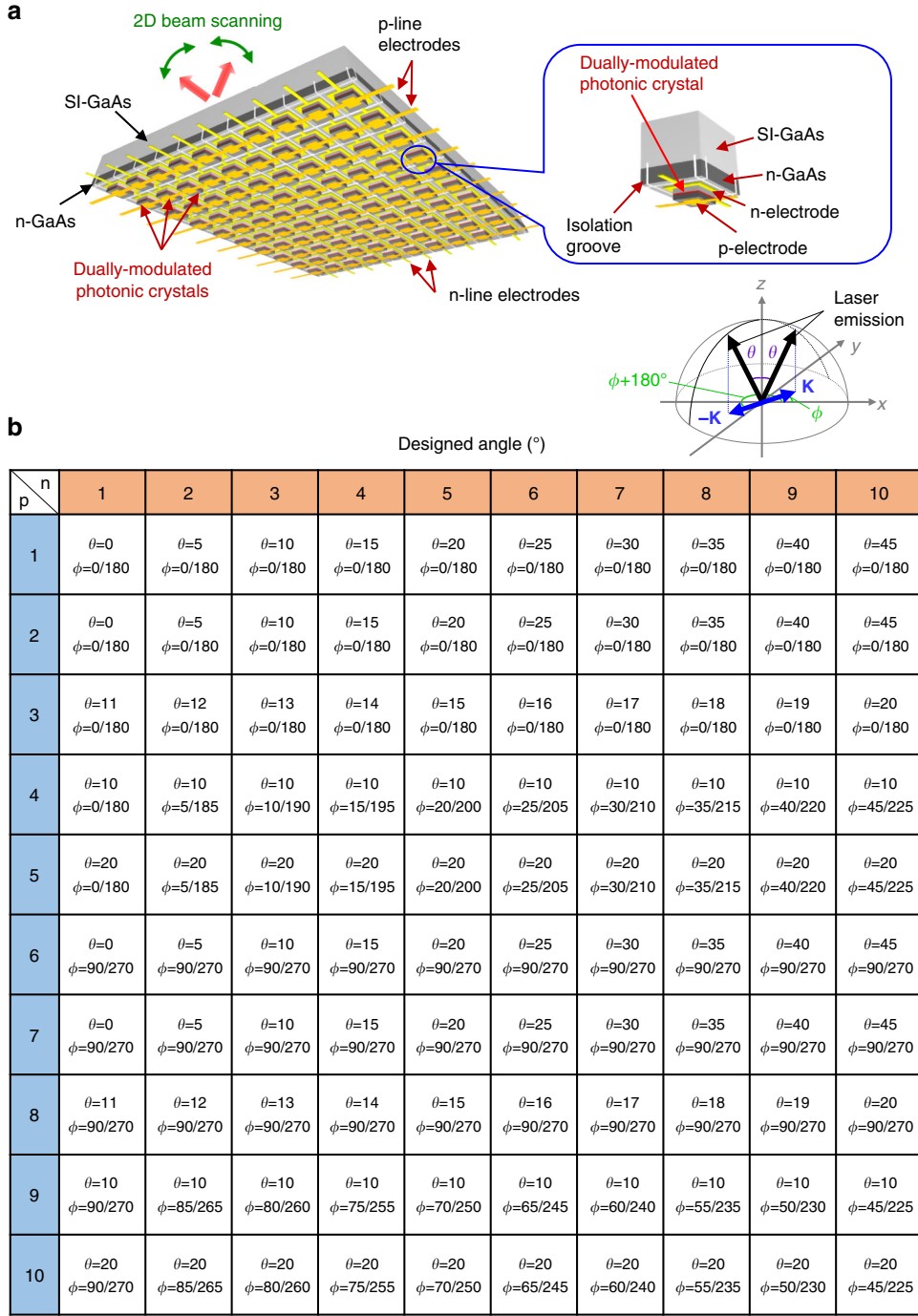

**Fig. 4 Design of an on-chip dually modulated photonic-crystal laser array. a** Schematic diagram of a dually modulated photonic-crystal laser array. Each lasing area is electrically isolated by isolation grooves. Lasers at each intersection of p- and n-line electrodes can be selectively operated by driving the corresponding line electrodes. **b** Designed angles ($\theta$ and $\phi$) of 100 laser elements (10 × 10) in the array. The emission angle definitions are shown in the inset.

the third row of Fig. 6a, we show snapshots of parallel 2D beam scanning, in which the scanning of polar angles (from $\theta = 0°$ to 45°) with different azimuthal angles ($\phi = 0°/180°$ and 90°/270°) was realized by driving two p-line electrodes (p-1 and p-6) and switching the n-line electrodes (from n-1 to n-10) concurrently. Meanwhile, the first row of Fig. 6b shows the scanning of azimuthal angles (from $\phi = 0°/180°$ to 45°/225°) with fixed polar angles $\theta = 20°$ by driving one p-line electrode (p-5) and switching the n-line electrodes (from n-1 to n-10), and the second row of Fig. 6b shows the scanning of polar angles (from $\theta = 0°$ to 45°)

with fixed azimuthal angles $\phi = 90°/270°$ by driving one p-line electrode (p-6) and switching the n-line electrodes (from n-1 to n-10). The third row of Fig. 6b shows parallel 2D beam scanning, where we realize the simultaneous scanning of both polar angles (from $\theta = 0°$ to 45°) and azimuthal angles (from $\phi = 0°/180°$ to 45°/225°). A video of 2D beam scanning performed in real time, for not only the range of angles displayed in Fig. 6, but also various other angles with single and parallel operation, is provided as a Supplementary Movie 1 and is described in Supplementary Note 6.

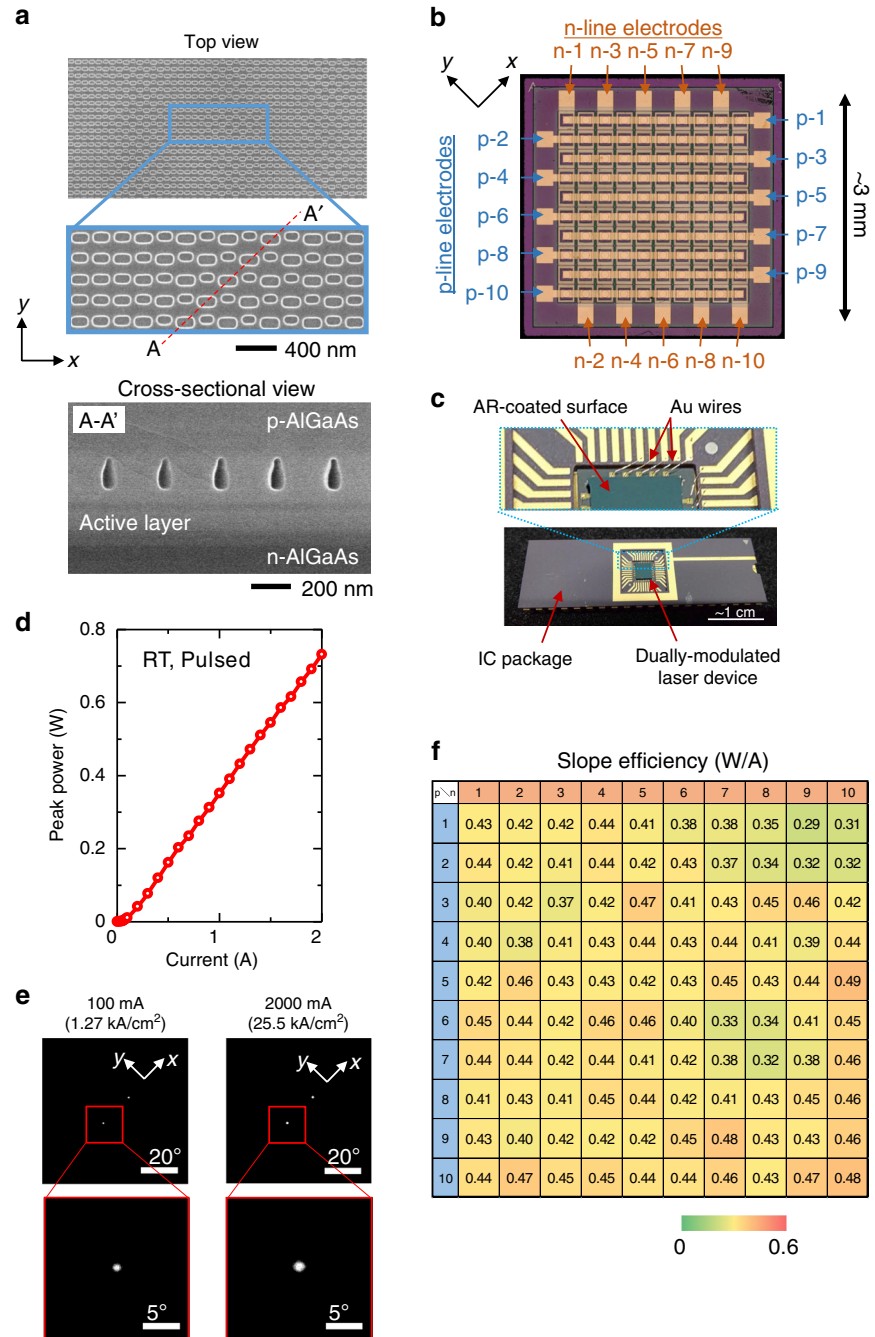

**Fig. 5 Fabrication and characterization of the on-chip dually modulated photonic-crystal laser array. a** Top and cross-sectional SEM images of a dually modulated photonic crystal before and after air-hole-retained regrowth by MOVPE, respectively. The photonic crystal is fabricated for emission at $\theta = 10°$ and $\phi = 0°/180°$. **b** Optical microscope image of the back side of the fabricated device, with electrodes clearly visible. **c** Device after having been flip-chip bonded to an IC package. Laser light emitted from the AR-coated surface of the SI-GaAs substrate. **d** Current light-output characteristics measured by driving electrodes n-1 and p-4 under pulsed conditions (100 ns, 1 kHz) at room temperature (RT). **e** Far-field patterns for emission at $\theta = 10°$ and $\phi = 0°/180°$ at different injected currents. **f** Measured slope efficiency of all 100 integrated lasers.

## Discussion

In this study, we have demonstrated beam scanning using dually modulated photonic-crystal lasers with 100 resolvable points. In the future, it is expected that the number of resolvable points can be greatly increased at the cost of a proportionally smaller increase of the device size. For example, the number of resolvable points can be increased by a factor of 900 (to 90,000) while increasing the device size by only a factor of 4, as detailed in Supplementary Note 7. Even in this case, each excitation area can

be maintained at $100 \times 100\ \mu m^2$, so a narrow divergence angle and a high, watt-class output power can be preserved.

We consider that the number of resolvable points, as well as divergence angles, output powers, and total device sizes can be tailored to meet application-specific targets. For example, a configuration with 100 resolvable points, similar to that demonstrated above, can be applied to a unique new LiDAR system, in which the benefits of flash-type[15] and beam-scanning-type ToF LiDAR systems are integrated (see Supplementary Note 8 for

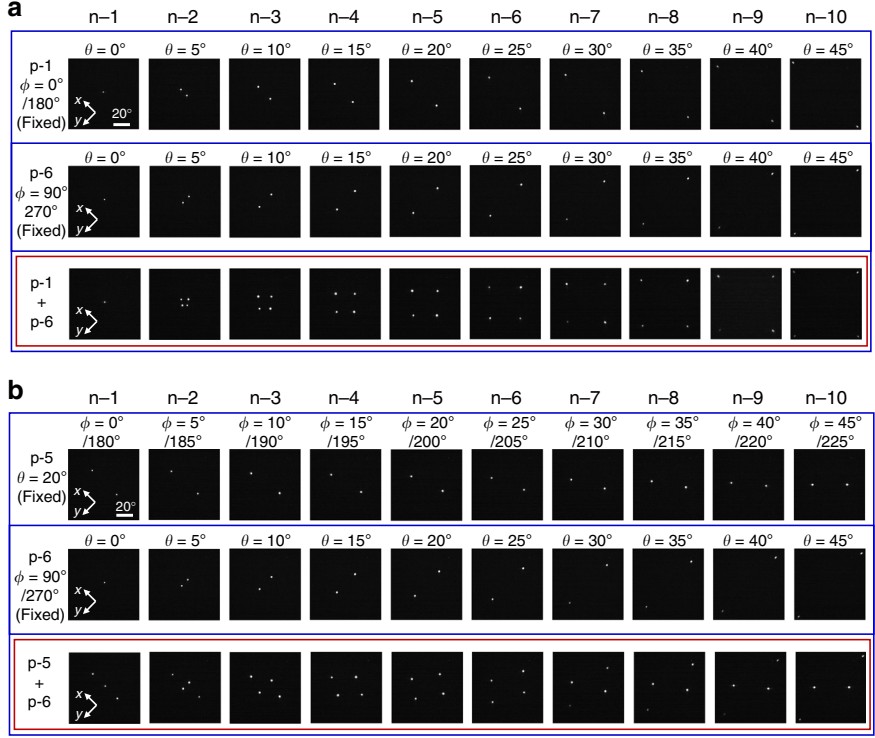

**Fig. 6 Demonstration of 2D beam scanning by selective driving of elements of the 10 × 10 matrix array. a** (Top two rows) Scanning polar angles (from $\theta = 0°$ to 45°) with fixed azimuthal angles ($\phi = 0°/180°$ or $90°/270°$). (Bottom row) Parallel 2D beam scanning of both top rows simultaneously. **b** (Top row) Scanning azimuthal angles (from $\phi = 0°/180°$ to $45°/225°$) with fixed polar angle $\theta = 20°$. (Middle row) Scanning polar angles (from $\theta = 0°$ to 45°) with fixed azimuthal angle $\phi = 90°/270°$. (Bottom row) Parallel 2D beam scanning of both top and middle rows simultaneously.

details). The parallel beam scanning described above would be useful for such a new LiDAR system.

We note that our device has an additional advantage in terms of robustness to temperature; the emission angle of our device changes only on the order of $10^{-4}$–$10^{-3}$ ° K$^{-1}$ in theory, which is one order of magnitude smaller than that of grating couplers (the latter being determined by the temperature dependence of the refractive index). The details are discussed in Supplementary Note 9. This robustness to temperature is attributed to the radiation wavevector **K**, which is fixed by the physical patterning of the dually modulated photonic crystals.

In conclusion, we have proposed the concept of dually modulated photonic crystals, and we have shown that the simultaneous modulation of both position and size of the photonic-crystal lattice points are essential to realize the emission of a beam in any 2D direction with a high-output power (watt-class, and potentially even higher) and high beam quality. We have fabricated an on-chip, circuit-driven array of 100 dually modulated photonic-crystal lasers, with which we have demonstrated the scanning of beams over a wide range of directions in two dimensions, as well as the simultaneous 2D scanning of multiple beams. It is expected that the device configuration can be modified so that number of resolvable points, divergence angle, output power, and total device size can meet application-specific targets. Our devices, with such excellent performance and novel functionalities, supply a variety of choices to the fields of sensing for autonomous driving and robot mobility, contributing to the progress of smart Society 5.0[16]. Our non-mechanical, high-power, high-beam-quality 2D beam-scanning technique would be also useful to other applications, such as adaptive illuminations.

## Data availability

The authors declare that the all the data supporting the findings of this study are available within this article and its Supplementary Information files, and are also available from the corresponding author upon reasonable request.

## Code availability

All associated code for 3D-CWT simulations are available from the corresponding author upon reasonable request.

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

## Acknowledgements
This work was carried out under the CREST program (JP MJCR17N3) commissioned by the Japan Science and Technology Agency (JST), Japan, and under the project of Council for Science, Technology and Innovation (CSTI), Cross ministerial Strategic Innovation Promotion Program (SIP), "Photonics and Quantum Technology for Society 5.0" (Funding agency: QST). We thank Kyoko Kitamura, Eiji Miyai, and Wataru Kunishi for fruitful discussions.

## Author contributions
S.N. planned and directed this work. R.S., K.I., and M.D.Z. carried out the detailed design of the modulated photonic crystals and fabricated the arrayed device with S.F and M.Y. R.S. and M.D.Z. measured the lasing characteristics with K.I and S.F. R.S., T.I., and Y.T. conducted the theoretical analysis as well as the formulation of nanoantenna theory with J.G. R.H. performed epitaxial growth for the device. S.N. and R.S. wrote the manuscript with T.I., K.I., M.D.Z., and J.G.

## Competing interests
The authors declare no competing interests.
