## [Peer Review File · Nature Communications]

REVIEWER COMMENTS

Reviewer #4 (Remarks to the Author):

I thank the authors for their careful consideration of our initial review. Their comments have clarified several aspects of the initial manuscript and have improved our understanding of the work's unique contributions. I would recommend publication in Nature Communications based on these updates. Below, I outline a few minor points that should be addressed.

The comparison of the dually modulated photonic crystal device with a grating-steered PCSEL certainly helps to address our questions regarding the differences between the two approaches. In the new supplemental section S2, the authors state that the diffraction efficiency into the first order mode is 60% at best ("The ratio of these unwanted diffractions/reflections to the incident power is over 40%, even for optimized rectangular gratings"). Is the 40% loss a fundamental bound? Does the diffraction efficiency improve at larger angles, where higher diffraction orders will be prohibited? Other works ([1] for example) seem to demonstrate high efficiency grating-based tuning of the first diffraction order, albeit in a different geometry than explored here.

The re-written section on 3D-CWT is now clearer. The authors explain that the updated simulations agree with the nano-antenna theory, and that certain designs ($\Delta d = 0.08a$ for all ΔS shown in Fig. S3b) yield high efficiency steering with little light trapped inside the device. However, our question from the initial review is still of interest: "What absolute emission angle do these parameters correspond to?" Does the loss due to total internal reflection increase at larger steering angles? It would be insightful to show Fig. S3b for a few other values of Δd used in the fabricated structures.

In response to the "emission homogeneity" questions in our initial review, the authors have provided new details on the beam quality statistics. Although the current manuscript targets time-of-flight LiDAR as an application where wavelength homogeneity may not be necessary, including the emission wavelength statistics of the fabricated devices could be helpful for future PhC-based beam steering devices.

Finally, it would be helpful to include a reference for the claim "the emission angle of our device changes only on the order of $10^{-4} \sim 10^{-3}$ °/K in theory, which is one order of magnitude smaller than that of grating couplers (the latter being determined by the temperature dependence of the refractive index)," as the conversion from the thermo-optic coefficient ($\sim 10^{-4}$ /K) to angle shift/temperature change is not obvious.

We thank the authors once again for their detailed responses and revisions.

1. Huang, Y.-W. W. et al. Gate-Tunable Conducting Oxide Metasurfaces. *Nano Lett.* 16, 5319–5325 (2016).

Reply to Reviewer #4

General Comment

I thank the authors for their careful consideration of our initial review. Their comments have clarified several aspects of the initial manuscript and have improved our understanding of the work's unique contributions. I would recommend publication in Nature Communications based on these updates. Below, I outline a few minor points that should be addressed.

Reply

We are grateful to the reviewer for carefully reading our manuscript and kindly recommending its publication in Nature Communications. The reviewer's words are greatly encouraging to us. Each of the reviewer's minor points are addressed below.

Comment 1

The comparison of the dually modulated photonic crystal device with a grating-steered PCSEL certainly helps to address our questions regarding the differences between the two approaches. In the new supplemental section S2, the authors state that the diffraction efficiency into the first order mode is 60% at best ("The ratio of these unwanted diffractions/reflections to the incident power is over 40%, even for optimized rectangular gratings"). Is the 40% loss a fundamental bound? Does the diffraction efficiency improve at larger angles, where higher diffraction orders will be prohibited? Other works ([1] for example) seem to demonstrate high efficiency grating-based tuning of the first diffraction order, albeit in a different geometry than explored here.

[1] Huang, Y.-W. W. et al. Gate-Tunable Conducting Oxide Metasurfaces. *Nano Lett.* 16, 5319–5325 (2016).

Reply

We are very happy to know that the reviewer kindly understands the difference between the two approaches. The answer to the rest of reviewer's concern is as follows: The diffraction efficiency of ~60% is the maximum transmittance (upper bound) for the rectangular gratings designed for the 1st order diffraction angles for $|\theta| < 30^\circ$. The calculated maximum transmittance by 1st-order diffraction for a wider range of angles is shown in the revised Fig.S4d (see below). From this figure, it is seen that the efficiencies can be slightly improved, as the reviewer points out, at larger angles ($|\theta| > 30^\circ$) where 2nd- and higher-order diffractions are prohibited. Be that as it may, the lower diffraction efficiency (<60%) at the angles of $|\theta| < 30^\circ$ nevertheless hinders the realization of highly efficient beam scanning over the entire wide field of view, which is required in many practical applications. Furthermore, in LiDAR applications in particular, the emission of superfluous beams generated by 0th-

2nd-, and higher-order diffractions will even induce noise in the reflected signal. Besides, even for the above optimized operation conditions ($|\theta| > 30^\circ$ and perpendicular polarization), a relatively large portion of the incident power ($\sim 20\%$) is still lost as unwanted 0th and 1st-order reflection by the grating. Such unwanted reflection may also induce noise and instability in the lasing mode. It should also be noted that the “reflective-type” gratings reported in the above reference [1] are difficult to be integrated on top of PCSELS (or VCSELS) for realizing one-chip beam-scanning devices. Therefore, our dually-modulated PCSELS, where 1st-order beams are exclusively emitted for all emission angles θ without using any additional elements, have fundamental merits over a combination of independently fabricated 2D grating and PCSELS (or VCSELS). We hope for the reviewer’s kind understanding of this point.

Revised Figure S4. (a) Schematics of beam diffraction with a rectangular grating on top of the vertically emitted PCSEL. (b), (c) Examples of calculated transmittance and reflectance for each beam diffracted by gratings designed to obtain desired beam directions of $|\theta| = 5^\circ$ and 10° , respectively, by $\pm 1^{\text{st}}$ order diffraction. (d) Calculated maximum transmittance in $\pm 1^{\text{st}}$ order diffraction as a function of the

diffraction angle $|\theta|$ when the polarization is perpendicular (red) and parallel (black) to the grooves of the grating.

In the revised version of Supplementary Section S2, we have included the above discussion, and also added new Fig.S4(d).

Comment 2

The re-written section on 3D-CWT is now clearer. The authors explain that the updated simulations agree with the nano-antenna theory, and that certain designs ($\Delta d=0.08a$ for all ΔS shown in Fig. S3b) yield high efficiency steering with little light trapped inside the device. However, our question from the initial review is still of interest: “What absolute emission angle do these parameters correspond to?” Does the loss due to total internal reflection increase at larger steering angles? It would be insightful to show Fig. S3b for a few other values used in the fabricated structures.

Reply

We thank the reviewer very much for these valuable comments. The absolute emission angle of Fig. S3b of first round of revised paper corresponded to $(\theta, \Phi) = (36^\circ, 0^\circ/180^\circ)$. We had confirmed that the loss due to total internal reflection does not increase even at larger steering angles. As the reviewer has suggested, we have added calculations for a few other steering angles of $\theta = 17^\circ, 28^\circ, \text{ and } 54^\circ$ for $\Phi = 0/180^\circ$ in the revised Fig. S3 (see the separately prepared, revised Supplementary Section S1).

Comment 3

In response to the “emission homogeneity” questions in our initial review, the authors have provided new details on the beam quality statistics. Although the current manuscript targets time-of-flight LiDAR as an application where wavelength homogeneity may not be necessary, including the emission wavelength statistics of the fabricated devices could be helpful for future PhC-based beam steering devices.

Reply

Thank you very much for this suggestion. All 100 laser elements have been designed to possess the same lasing wavelength, though with different emission directions. As already described in the first round of the revised paper, the lasing wavelengths are around 940nm (specifically, $\sim 949\text{nm}$ when the oscillation occurs in the mode of band edge A or B, and $\sim 932\text{nm}$ when in the mode of band edge

C or D). We have evaluated the standard deviation of these lasing wavelengths (for all 100 samples), and found that the standard deviation from the mean wavelength is as small as around ~ 0.2 nm, which is within the resolution limit of our spectrometer. The above information has been added to the revised manuscript from line 303 to line 306.

Comment 4

Finally, it would be helpful to include a reference for the claim “the emission angle of our device changes only on the order of 10^{-4} – 10^{-3} °/K in theory, which is one order of magnitude smaller than that of grating couplers (the latter being determined by the temperature dependence of the refractive index),” as the conversion from the thermo-optic coefficient ($\sim 10^{-4}$ /K) to angle shift/temperature change is not obvious.

Reply

Thank you very much for this valuable comment. In response, we have added a derivation of the emission angle change with respect to temperature, to Supplementary Section S9, and have slightly modified the manuscript (lines 307, 367 and 368) to reflect this addition.

Supplementary Information

§ S9. Estimation of emission angle change with respect to temperature

In this section, we calculate the emission angle change of grating couplers and dually modulated photonic crystal lasers with respect to temperature.

Grating couplers

First, we discuss the emission angle change of grating couplers with respect to temperature. The emission angle θ_g of a grating coupler at a given temperature is given by equation S39 [S6].

$$\sin\theta_g = \frac{\Lambda n_{\text{eff}} - \lambda}{\Lambda} \quad (\text{S39})$$

Here, Λ is the grating period, n_{eff} is the effective refractive index of the grating, and λ is the wavelength. When the temperature of the grating changes, the grating period and the effective refractive index change, as well. On the other hand, for the common case in which an external laser source is used, the wavelength remains constant. Thus, the change of emission angle with respect to the temperature T is given by:

$$\frac{\Delta\theta_g}{\Delta T} = \frac{1}{\cos\theta_g} \left(\frac{\Delta n_{\text{eff}}}{\Delta T} + \frac{\lambda}{\Lambda} \frac{(\Delta\Lambda/\Lambda)}{\Delta T} \right) \quad (\text{S40})$$

The grating period changes due to thermal expansion of the semiconductor material used for the gratings. For example, the thermal expansion coefficient of silicon is $2.6 \times 10^{-6} / \text{K}$. Thus, $(\Delta A/A)/\Delta T = 2.6 \times 10^{-6} / \text{K}$ for silicon gratings. Meanwhile, the thermo-optic coefficient $\Delta n/\Delta T$ of silicon is $\sim 2 \times 10^{-4} / \text{K}$ [S7]. Assuming the effective refractive index of the grating coupler is ~ 3 , $\Delta n_{\text{eff}}/\Delta T \approx 1.5 \times 10^{-4} / \text{K}$, which is two orders of magnitude larger than $(\Delta A/A)/\Delta T$. Thus, the change of emission angle mostly depends on the change of refractive index for silicon gratings.

$$\frac{\Delta \theta_g}{\Delta T} \approx \frac{1}{\cos \theta_g} \left(\frac{\Delta n_{\text{eff}}}{\Delta T} \right) \quad (\text{S41})$$

Given an emission angle of 10° , $\Delta \theta_g/\Delta T$ is calculated to be $\sim 9 \times 10^{-3} / \text{K}$.

Dually modulated photonic crystal lasers

Next we discuss the emission angle change of dually modulated photonic crystal lasers with respect to temperature. As mentioned in the manuscript, the vector \mathbf{K} determines the emission direction in free space. In terms of polar emission angle θ and azimuthal emission angle Φ :

$$\mathbf{K} = \frac{2\pi}{\lambda} (\sin \theta \cos \Phi, \sin \theta \sin \Phi) \quad (\text{S42})$$

$$|\mathbf{K}| = \frac{2\pi}{\lambda} \sin \theta$$

\mathbf{K} is formed by modulating the photonic crystal lattice points, and its magnitude is inversely proportional to the photonic crystal lattice constant a (i.e., $|\mathbf{K}| \propto 1/a$). As with the silicon grating couplers discussed above, the effect of thermal expansion (i.e., change of a) of the gallium arsenide photonic crystal is two orders of magnitude smaller than its refractive index change. Therefore, $|\mathbf{K}|$ is considered to remain constant with temperature. However, unlike with grating couplers, the change of refractive index with temperature now affects the emission wavelength λ . This implies that the polar emission angle changes by:

$$|\mathbf{K}| \frac{\Delta \lambda}{\Delta T} \approx 2\pi \cos \theta \frac{\Delta \theta}{\Delta T} \quad (\text{S43})$$

Substituting $|\mathbf{K}|$ in equation S42 into equation S43, we obtain:

$$\frac{\Delta \theta}{\Delta T} \approx \tan \theta \frac{1}{\lambda} \frac{\Delta \lambda}{\Delta T} \quad (\text{S44})$$

The change of the emission wavelength of our photonic crystal lasers with respect to temperature ($\Delta \lambda/\Delta T$) is found in experiments to be 0.086 nm/K . Thus, given the same emission angle of 10° as above, $\Delta \theta/\Delta T$ is calculated to be $\sim 9 \times 10^{-4} / \text{K}$. This is one order of magnitude smaller than the grating couplers.

- S6. Hulme, J.-C., Doylend, J.-K., Heck M.-J.-R., Peters, J.-D., Davenport, M.-L., Bovington, J.-T., Coldren, L.-A., and Bowers, J.-E., *Optics Express*, **23**, 5861 (2015).
- S7. Komma, J., Schwarz, C., Hofmann, G., Heinert, D., and Nawrodt, R., *Appl. Phys. Lett.* **101**, 041905 (2012).